# A Narrative Review of Salivary Gland Extracellular Matrix and Sjögren’s Syndrome: Research Status and Future Prospects

**DOI:** 10.3390/biom16010072

**Published:** 2026-01-01

**Authors:** Haodong Su, Xinying Fan, Chunxia Nie, Xiaoyu Tang, Jingjin Hu, Ke Xu, Liyun Zhang, Dan Ma

**Affiliations:** 1Third Hospital of Shanxi Medical University, Shanxi Bethune Hospital, Shanxi Academy of Medical Sciences, Tongji Shanxi Hospital, Taiyuan 030000, China; suhaodong@sxmu.edu.cn (H.S.); fanxinying1@sxmu.edu.cn (X.F.); niechunxia@sxbqeh.com.cn (C.N.); polaris@sxmu.edu.cn (X.T.); natural@sxmu.edu.cn (J.H.); xuke@sxbqeh.com.cn (K.X.); zhangly@sxbqeh.com.cn (L.Z.); 2Shanxi Province Clinical Research Center for Dermatologic and Immunologic Diseases (Rheumatic Diseases), Taiyuan 030000, China; 3Shanxi Province Clinical Theranostics Technology Innovation Center for Immunologic and Rheumatic Diseases, Taiyuan 030000, China; 4Shanxi Academy of Advanced Research and Innovation, Taiyuan 030000, China

**Keywords:** Sjögren’s syndrome, salivary gland, extracellular matrix, matrix metalloproteinases, fibrosis

## Abstract

Sjögren’s syndrome (SS) is a chronic systemic autoimmune disease characterized by the main clinical manifestation of oral and ocular dryness, predominantly affecting middle-aged and elderly women. As the most commonly affected target organs in SS, pathological changes in the salivary glands (SGs) and their underlying mechanisms are of great significance for understanding the disease progression. Recent studies have revealed that a dynamic imbalance of the extracellular matrix (ECM) in the SGs plays a crucial role in the pathogenesis of SS. Dysregulation of matrix metalloproteinases (MMPs) and the fibrotic processes they mediate constitute the core pathological changes. These alterations intertwine with local chronic inflammatory responses, cellular senescence, and hyperosmolarity, collectively leading to the destruction of the SG parenchymal structure and progressive loss of secretory function, significantly impairing the patients’ quality of life. However, research on the pathological mechanisms of the SG ECM remains insufficient, and there are currently no specific therapeutic interventions targeting ECM alterations in clinical practice. This review systematically elucidates the characteristics of pathological and physiological changes in the SG ECM in SS and thoroughly explores novel therapeutic strategies based on ECM regulation, as well as their clinical application prospects.

## 1. Introduction

Sjögren’s syndrome (SS) is a chronic inflammatory autoimmune disease characterized by dry mouth and eyes, and is often accompanied by systemic damage to the digestive system, lungs, kidneys, skin, and joints. The disease predominantly affects middle-aged and elderly women and is characterized by impaired exocrine gland function, lymphocyte infiltration, and production of various autoantibodies [1]. The exact pathogenesis of SS remains unclear. Current treatment strategies primarily rely on artificial saliva and tears to alleviate dryness, along with corticosteroids, disease-modifying antirheumatic drugs, and biologics to suppress inflammation and modulate the immune function. However, some patients respond poorly to existing treatments, and the long-term use of immunosuppressants may increase the risk of infection. Therefore, a deeper understanding of the pathogenesis of SS and the development of new therapeutic targets are crucial for improving patient outcomes.

The SGs serve as key sites of lymphocyte infiltration during the pathogenesis of SS, where their ECM interacts with tissue cells to sustain the normal structure and function of the gland. Current research on SG dysfunction has focused primarily on specific cell types such as lymphocytes and salivary gland epithelial cells (SGECs). However, changes in the cell function are often accompanied by significant ECM remodeling. ECM, which maintains tissue integrity, homeostasis, and repair, plays a critical role in SS. Dysregulation of the ECM in SS leads to an abnormal matrix composition and tissue structure, affecting cell activity and glandular function. Therefore, understanding the pathological changes in SG ECM in SS is essential for uncovering disease mechanisms and exploring new therapeutic directions. This article aims to systematically review the pathological characteristics of the ECM in SGs in SS and explore potential therapeutic strategies based on ECM repair, with the goal of providing theoretical evidence and translational insights to improve the prognosis of patients with SS.

## 2. Composition and Function of SG ECM

The SG ECM is a three-dimensional network synthesized and secreted by cells and is primarily composed of proteins and polysaccharides [2]. The SG ECM is divided into two types: the interstitial matrix and the basement membrane. The interstitial matrix surrounds the tissue cells and provides structural support, whereas the basement membrane is a specialized ECM that separates epithelial cells from the surrounding matrix [3]. The interstitial matrix includes glycosaminoglycans, proteoglycans, collagen, elastin, and non-collagenous glycoproteins, such as fibronectin and laminin [3]. Glycosaminoglycans and proteoglycans form a gel-like matrix that provides tissue elasticity [4], while collagen and elastin form the structural framework of the ECM [5,6]. Fibronectin and laminin, as adhesion molecules, regulate cell survival, proliferation, differentiation, adhesion, and migration by interacting with various cells and ECM components [7,8]. The basement membrane is primarily composed of type IV collagen, laminin, nidogens, and various proteoglycans. It provides structural support for epithelial tissues, connects epithelial and connective tissues, and acts as a selective barrier that regulates molecular permeability and cell movement [9]. The basement membrane also plays a crucial role in maintaining the polarity of SGECs and coordinating cell-ECM signaling [10]. Additionally, the ECM regulates immune cell activity, migration, and localization, maintains the overall stability, and restores integrity after injury [11]. The ECM and tissue cells are interdependent. Cells regulate ECM synthesis and degradation, whereas the ECM provides support, protection, and nutrition to cells, regulating their proliferation, differentiation, metabolism, recognition, adhesion, and migration [2].

## 3. Pathological Changes in SG ECM in Sjögren’s Syndrome

The composition and distribution of the ECM undergo dynamic changes over time. During early embryonic development, ECM remodeling is closely related to the branching morphology of the SGs [12]. As the SG matures, the ECM components are continuously renewed to maintain normal gland structure and function. However, during aging, collagen deposition increases in the SGs, accelerating fibrosis and leading to progressive gland dysfunction [13]. Although fibrosis is a natural outcome of SG aging, SS significantly accelerates this process [14]. Dysregulation of matrix metalloproteinases (MMPs) and SG fibrosis are key events in SS that disrupt ECM homeostasis and exacerbate glandular dysfunction (Figure 1).

### 3.1. Dysregulation of SG MMPs

Under homeostatic conditions, ECM remodeling is a tightly regulated process. Tissue cells continuously synthesize and secrete ECM components to maintain normal physiological functions, and regulate ECM degradation by secreting MMPs, their endogenous inhibitors, and tissue inhibitors of metalloproteinases (TIMPs). This balance is crucial for wound healing, tissue remodeling, and cell migration [15]. MMPs are a family of zinc-dependent endopeptidases that degrade ECM components, facilitate ECM turnover, and maintain tissue structure and function. MMP activity is strictly regulated by TIMP, and this dynamic balance plays a key role in various physiological processes [16]. In SG tissues, MMPs are primarily synthesized and secreted by SGECs. Studies have shown that MMP-2, MMP-3, MMP-9, MMP-14, MMP-15, TIMP-1, TIMP-2, and TIMP-3 are involved in maintaining the ECM homeostasis in SGs [17,18,19]. When MMP activity is unbalanced, excessive ECM degradation leads to the destruction of tissue structure, causing various types of SG damage. Chronic tissue damage not only results in the deposition of harmful substances but also triggers abnormal tissue repair responses, ultimately driving fibrosis [17]. This pathological ECM remodeling disrupts the dynamic balance and severely affects SG function.

#### 3.1.1. Inflammation-Induced MMP Dysregulation

Under the influence of infection and environmental factors, SS patients exhibit increased SGEC apoptosis, expressing various pattern recognition receptors (PRRs) that activate the innate immune system and promote T- and B-cell activation and differentiation, thereby triggering adaptive immune responses. Immune activation leads to the production of numerous inflammatory factors, immunoglobulins, and autoantibodies. They also express various cytokine receptors and secrete inflammatory factors, including interferon (IFN), transforming growth factor (TGF), interleukin (IL)-1, IL-6, and B-cell activating factor (BAFF). Additionally, SGECs secrete chemokines such as CXCL8 and CXCL10, which attract inflammatory cells to the gland and exacerbate local inflammation [20]. Studies have shown that patients with SS have increased levels of hyaluronic acid (HA) in their saliva [21]. HA, as an ECM component, can covalently modify the heavy chain (HC) of the inter-alpha-inhibitor (IαI) family under inflammatory conditions, increasing its affinity for CD44 and binding to immune cells, prolonging inflammation in the ECM of diseased tissue [22,23,24].

MMPs in SGs are primarily produced by SGECs at low levels under steady-state conditions. However, MMP transcription is significantly upregulated upon stimulation with chemokines, cytokines, or growth factors. This upregulation leads to extensive ECM remodeling, causing imbalances in ECM components and driving disease progression [25]. Immune cells are a significant source of MMPs. For example, activated macrophages secrete membrane-type 1 matrix metalloproteinase (MT1-MMP) to degrade basement membrane components for migration [26], whereas neutrophils secrete MMP-9 to enter infected sites [27]. Studies have found that, in patients with SS, the expression and activity of MMP-3 and MMP-9 are significantly increased in the SG tissue, with the degree of increase correlating with disease severity and extensive ECM remodeling [28]. The extent of SG damage is closely related to an increase in MMP-9/TIMP-1 and MMP-3/TIMP-1 ratios [29]. Therefore, inflammation-induced MMP dysregulation leads to SG tissue damage in SS, which plays a key role in the disease pathology.

#### 3.1.2. Senescence-Induced MMP Dysregulation

Cellular senescence involves significant biological changes including morphological alterations, chromatin remodeling, and metabolic reprogramming. Senescent cells secrete a series of pro-inflammatory factors, known as the senescence-associated secretory phenotype (SASP), including IL-1, IL-6, IL-7, IL-8, IL-18, and tumor necrosis factor (TNF)-α. The release of these factors can cause low-level chronic inflammation in the tissues and accelerate organ dysfunction [30,31,32,33]. Studies have shown that in aging mouse models, SG tissue exhibits significant inflammatory cell infiltration and enhanced immune responses, with elevated levels of inflammatory factors such as TNF-α, TGF-β1, and IL-6, leading to MMP dysregulation and affecting SG structure and function [34,35]. An animal study further demonstrated that aged mice (24 months old) had significantly higher levels of MMP-2 and type I collagen in their SGs than young mice (3 months old) [36]. Another study showed that in 18-month-old mice, the immunoreactivity of MMP-2 in SG tissue was significantly higher than that in 2-month-old mice, accompanied by reduced acinar cell numbers, increased eosinophilic zymogen granules, collagen accumulation in fibrotic areas, and interlobular duct dilation [37]. These findings suggest that aging may affect the MMP expression and activity, leading to pathological changes in SG tissues. Additionally, Nassar et al. reported age-related changes in salivary biomarkers in healthy individuals. Compared with younger subjects, older subjects had significantly reduced salivary flow rates and increased concentrations of MMP-1 and MMP-8 in their saliva [38]. This further supports the association between aging and MMP dysregulation, indicating that aging may disrupt the MMP balance and impair normal SG structure and function.

Notably, the SG tissue of patients with SS exhibits chronic inflammatory stimulation and reactive oxygen species (ROS) accumulation [20,39]. Studies have shown that inflammation-mediated immune system overactivation and ROS-induced telomere damage are key mechanisms that promote cellular senescence [30,40]. Therefore, in patients with SS, chronic inflammation and ROS in the SG tissue may play a crucial role in cellular senescence. However, direct evidence elucidating the interactions among inflammation, ROS, and cellular senescence in SS and their specific effects on MMP dysregulation is lacking. Therefore, further research is required in this area.

#### 3.1.3. Hyperosmolarity-Induced MMP Dysregulation

In patients with SS, the SG tissue exhibits increased acinar cell apoptosis [41] and altered expression or distribution of aquaporins (AQP) 4 and AQP5 [42,43,44]. These changes lead to increased local osmotic pressure in the SGs, creating a hyperosmolar state. In vitro studies have shown that chronic hyperosmolar stimulation can induce SGECs to upregulate IFN signaling pathway-related genes, IL-17 signaling pathway-related genes, and various chemokine genes, thereby promoting T-cell- and B-cell infiltration and significantly increasing the expression of MMP-1, MMP-3, MMP-13, and MMP-25 [45]. These findings suggest that local hyperosmolarity may regulate MMP expression and activity, and participate in ECM remodeling. Therefore, in patients with SS, hyperosmolarity in the SG tissue may be a significant factor contributing to MMP dysregulation. This discovery not only reveals the impact of hyperosmolar environments on immune responses and ECM remodeling in SS but also provides new research directions for further exploration of the pathogenesis of SS.

#### 3.1.4. Epigenetic Regulation-Mediated MMP Dysregulation

Recent studies have shown that epigenetic mechanisms play a critical role in the transcriptional regulation of MMP and may potentially drive the pathological progression of SS. Epigenetic regulation refers to the process of controlling gene expression through mechanisms such as DNA methylation, histone modification, chromatin remodeling, and non-coding RNAs without altering the DNA sequence. Current research on epigenetic regulation in the SGs of SS patients remains relatively limited. This section focuses on elucidating the impact of DNA methylation and histone modifications on the dysregulation of MMP expression.

DNA methylation, one of the most representative epigenetic regulatory mechanisms, refers to the process in which a methyl group is added to the 5th carbon position of cytosine residues catalyzed by DNA methyltransferases (DNMTs), forming 5-methylcytosine. In SS, abnormal DNA methylation exhibits an established causal link with dysregulated MMP expression. Studies reveal that global DNA methylation levels are reduced in SG tissues from SS patients, accompanied by decreased DNMT activity. This hypomethylation state may lead to promoter demethylation of genes such as MMP-3 and MMP-9, consequently promoting their overexpression [46,47].

Histone modification refers to the chemical alteration processes of histones—including methylation, acetylation, phosphorylation, adenylation, ubiquitination, and ADP-ribosylation—catalyzed by specific enzymes. Among these, histone acetylation and deacetylation represent the most extensively studied modifications, which bidirectionally regulate gene transcription by altering histone charge states and chromatin structure. Histone acetyltransferases (HATs) transfer acetyl groups to lysine residues on histones, neutralizing their positive charge and reducing affinity for negatively charged DNA. This promotes chromatin relaxation and openness, thereby facilitating transcription factor binding and activation of downstream genes. Conversely, histone deacetylases (HDACs) suppress gene transcription by removing acetyl groups. Researchers commonly investigate histone acetylation’s role in inflammation-induced MMP expression using HAT inhibitors and HDAC inhibitors (HDACis). For instance, in preclinical models of experimental autoimmune encephalomyelitis, HDACi MS-275 (Entinostat) significantly reduced MMP-9 expression [48]. Additionally, IL-1β stimulation enhances histone H4 acetylation at AP-1-specific promoter sites of the MMP-1 and MMP-13 genes [49], an effect blocked by HDACi trichostatin A and sodium butyrate [50]. These findings suggest that both the local immune microenvironment in SS SGs and imbalanced HAT/HDAC activity may collectively exacerbate MMP dysregulation [51].

Research on epigenetic regulation-mediated MMP dysregulation mechanisms in SGs remains unexplored. Elucidating this mechanism will not only clarify the pathological basis of SS but also provide crucial direction for developing novel diagnostic biomarkers and precise therapeutic targets.

#### 3.1.5. Impact of MMP Dysregulation on SG Structure and Function

MMPs disrupt the normal SG structure and function through various mechanisms in SS. MMPs degrade key ECM components that directly damage the SG structure. Under normal conditions, cells connect to the basement membrane through integrins and dystroglycan receptors, which interact with actin and intermediate filaments in the cytoskeleton via focal adhesions and hemidesmosomes, thereby maintaining the integrity of the organelles, cell volume, structure, and apical microvilli [52,53]. MMPs degrade various basement membrane and ECM components, including laminin, fibronectin, collagen, and proteoglycans [54]. This degradation leads to the separation of acinar cells from the basement membrane, loss of cytoskeletal organization, acinar lumen dilation, loss of nuclear polarity, and microvillus disorganization [54]. In acinar cells, secretory granules fuse with the apical plasma membrane and release their contents into the acinar lumen to complete saliva secretion [55]. However, MMP-induced structural disruption interferes with this process, causing secretory granules to accumulate in acinar cells and fail to be released into the lumen, ultimately leading to a significant reduction in basal salivary flow rates [54]. Secretory dysfunction is a key pathological basis for dry mouth symptoms in patients with SS (Figure 2).

Second, MMP-degraded ECM products can act as damage-associated molecular patterns (DAMPs) by interacting with PRRs. Through these interactions, endogenous DAMPs trigger sterile inflammation, promote the production of inflammatory mediators, recruit immune cells to the site of injury, and serve as a bridge between innate and adaptive immunity [56]. For example, decorin, a typical ECM-derived DAMP, is significantly upregulated in the labial glands of patients with SS. It can activate PRRs to promote TNF-α production and induce SGECs apoptosis. Decorin also acts on macrophages, promoting their transition from a resting state (M0) to a pro-inflammatory state (M1) and secreting various pro-inflammatory factors such as TNF-α, IL-1β, and IL-6 [57]. Additionally, various cytokines and ECM degradation products can induce MMP expression, such as IL-1α, IL-6, TNF-α, IFN-γ, fibronectin, and laminin degradation fragments [58,59], creating a vicious cycle that exacerbates glandular damage.

Therefore, studying the role of MMP dysregulation in the pathogenesis of SS is crucial for SS treatment. Inhibiting MMP overexpression or blocking its pro-inflammatory effects may effectively alleviate the glandular damage and dysfunction in patients with SS. In addition, exploring the interactions between ECM degradation products and the immune system could help in the development of novel immunomodulatory therapies.

### 3.2. SG Fibrosis

Fibrosis is the result of an imbalanced or excessive repair response of the interstitial fibrous connective tissue to damaged parenchymal cells. Its main pathological feature is the increase in fibrous connective tissue and the reduction in parenchymal cells within the organ. In SS, chronic inflammation, cellular senescence, and a hyperosmolar environment can lead to MMP dysregulation. Both MMP-mediated degradation of the SG ECM and the local autoimmune response disrupt the normal architecture of epithelial cells and the basement membrane. This tissue damage initiates repair processes; however, chronic and persistent injury signals cause uncontrolled repair, ultimately resulting in the excessive deposition of collagen-dominated scar tissue, which replaces the functional acinar and ductal structures and thereby drives the progression of fibrosis [60]. SG biopsies show that the degree of fibrosis in SS patients is increased, positively correlated with lesion scores, and negatively correlated with the basal salivary flow rate [61,62]. In healthy SGs, the ECM is primarily composed of proteins such as collagen, laminin, fibronectin, and complex carbohydrates like glycosaminoglycans and proteoglycans [63]. However, during fibrosis, the expression and distribution of these ECM components become disordered, leading to the degeneration of acinar and ductal structures and functions [64].

Epithelial–mesenchymal transition (EMT) is a key mechanism in SG fibrosis. Studies have found that increased inflammation in the SG tissue of patients with SS is negatively correlated with the expression of epithelial markers (e.g., E-cadherin) and positively correlated with mesenchymal markers (e.g., vimentin and type I collagen) [65]. Additionally, immune cell infiltration and SGECs damage in the SGs of patients with SS lead to the production of various cytokines that play important roles in EMT. Among these, TGF-β is particularly crucial [66]. TGF-β has three subtypes: I, II, and III, with TGF-β1 being the primary mediator of fibrosis, often associated with the chronic phase of inflammatory diseases and leading to significant fibrotic changes that impair normal organ function [67]. In SS patients’ SGs, TGF-β1 expression is significantly increased and can induce EMT-dependent fibrosis through the TGF-β1/SMAD/Snail signaling pathway [68]. IL-13, IL-17, and IL-22 play important roles in fibrosis of various organs [69,70,71,72,73]. These cytokines are elevated in the SGs of patients with SS, with IL-17 and IL-22 promoting EMT by downregulating E-cadherin and upregulating vimentin and type I collagen expression [65,74]. The role of IL-13 is well-established in liver and lung fibrosis, where it can directly upregulate the expression of genes encoding pro-fibrotic components such as collagen [75,76]. Although direct evidence for its role in SG fibrosis in SS remains limited, elevated levels of IL-13 observed in SS patients suggest its potential involvement in this process [77]. Additionally, through the process of EMT, polarized epithelial cells gradually lose their apical-basal polarity and stable intercellular connections, acquiring a fibroblast-like phenotype and engaging in excessive secretion of collagen fibers. This ultimately leads to the destruction of SG structure and impairment of secretory function [78].

Fibrosis is a common endpoint in various chronic inflammatory diseases. Immune cells not only contribute to the pathogenesis of SS but also drive the progression of fibrosis. Studies have shown that nearly all immune cells are involved in organ fibrosis, with macrophages playing a particularly critical role in tissue repair and fibrotic programming. Tissue injury often promotes the polarization of macrophages towards a pro-inflammatory M2 phenotype, releasing cytokines such as IFN-γ, IL-4, IL-13, IL-10, and TGF-β. These cytokines subsequently promote EMT, regulate the activation of fibroblasts and myofibroblasts, stimulate ECM synthesis, and recruit inflammatory cells, thereby driving the onset of SG fibrosis [79,80]. Furthermore, fibrosis increases ECM stiffness and reduces elasticity, leading to a decreased activation threshold for TGF-β1 and the promotion of macrophage-mediated inflammatory responses, which exacerbate tissue fibrosis and create a vicious cycle [81,82]. T cells and B cells primarily promote SG fibrosis by mediating autoimmune responses, which trigger local chronic inflammation, SG damage, and macrophage activation [25,83]. Therefore, immune cell infiltration and the overexpression of related inflammatory factors are central to driving SG fibrosis. Targeting these factors and their signaling pathways may represent a potential therapeutic strategy to inhibit fibrosis and ameliorate the condition of SS.

It is noteworthy that the pathological process of SG fibrosis in SS exhibits similarities with another organ-specific autoimmune disease—Primary biliary cholangitis (PBC). PBC specifically targets the intrahepatic small bile duct epithelial cells, and its pathological features—chronic inflammation, bile duct destruction, and progressive liver fibrosis—closely mirror the damage to SG ducts and acini and the fibrotic patterns observed in SS. Consequently, some scholars suggest that PBC may be considered “Sjögren’s syndrome of the liver”, while SS could be analogized as “PBC of the salivary glands” [84]. This similarity extends beyond histology to immunological mechanisms. Both diseases share overlapping autoantibody profiles, with anti-SSA/Ro52 antibodies frequently detected in PBC patients. Notably, studies have shown that the presence of anti-Ro52 antibodies is associated with more advanced histological stages and greater fibrosis severity in PBC [85]. This suggests that in both SS and PBC, the immune response against common autoantigens like Ro52 may not only serve as a disease activity marker but could also be a key effector mechanism driving the progression from chronic tissue damage to fibrosis. Therefore, an in-depth investigation into the shared fibrotic pathways in SS and PBC may provide new perspectives for SS treatment.

## 4. Potential Therapeutic Approaches for Repairing SG ECM

Recent advances in the understanding of the pathophysiological changes in the SG ECM in SS have revealed potential therapeutic strategies that target key biomolecules and signaling pathways. In addition, the rapid development of stem cell and tissue engineering therapies offers promising prospects for ECM repair. Although current research on SG ECM repair remains limited, with no actively ongoing clinical studies, therapeutic strategies applied to fibrosis and autoimmune diseases provide valuable insights into SS treatment (Table 1).

### 4.1. Targeting Matrix Metalloproteinases

Early MMP inhibitors were broad-spectrum inhibitors that lacked selectivity and inhibited multiple MMP subtypes involved in both pathological and physiological processes. This nonselective inhibition disrupts normal ECM remodeling, causing tissue damage and adverse effects, thus limiting its clinical application [141,142]. As research on MMP biological functions has advanced, efforts have shifted toward developing highly selective, low-toxicity MMP inhibitors, particularly those targeting specific MMP subtypes.

In patients, MMP-3 and MMP-9 expression and activity are significantly increased and highly selective drugs that target these subtypes are currently under development. Chin et al. demonstrated that harmine, a specific MMP-3 inhibitor, ameliorated MMP-3-driven malignancies and inflammatory diseases [86]. This highly selective inhibitor specifically targets MMP-3, avoiding interference with other MMP subtypes. Unfortunately, the clinical application of this natural alkaloid has been impeded till now by severe toxic side effects, especially neurotoxicity, besides its poor water solubility [87]. Raeeszadeh-Sarmazdeh et al. designed a highly selective TIMP-1 variant that strongly inhibited MMP-3 and significantly reduced the inhibition of MMP-9, MMP-10, and other subtypes [143]. For MMP-9, andecaliximab, an MMP-9 inhibitor, showed good safety in a phase 1b clinical trial for patients with rheumatoid arthritis [88]; however, a subsequent efficacy evaluation study was terminated owing to limited patient enrollment (NCT02862574). In clinical studies for Crohn’s disease and advanced gastric cancer, andecaliximab demonstrated no significant difference in therapeutic effect compared to the placebo group [89,90].Additionally, Scannevin et al. found that the highly selective compound JNJ0966 effectively inhibited MMP-9 proenzyme activation and catalytic enzyme generation without significantly affecting other MMP subtypes (e.g., MMP-1, MMP-2, MMP-3, and MMP-14). In a mouse model of autoimmune encephalomyelitis, JNJ0966 significantly reduced the disease severity [78]. Peng et al. demonstrated that the selective small-molecule MMP-9 inhibitor (R)-ND-336 showed excellent efficacy in a diabetic mouse wound healing model, reducing macrophage infiltration and inflammation by inhibiting the MMP-9 activity and promoting wound healing [91].

Although these studies provide a theoretical basis and experimental support for the use of MMP inhibitors in the treatment of SS, no clinical trial data for patients with SS are available. Therefore, further research on the pathophysiological processes of the SG ECM in SS will provide more scientific evidence and therapeutic targets for MMP-targeted strategies. Additional animal and clinical studies are needed to confirm the safety and efficacy of these novel MMP inhibitors in SS.

### 4.2. Targeting TGF-β

Given TGF-β’s role in fibrosis, autoimmune diseases, and tumor development, it has garnered significant attention [144]. In SS patients’ SGs, increased TGF-β1 expression is a key mediator of EMT-dependent fibrosis [68], making TGF-β a potential therapeutic target. However, as a negative regulator maintaining immune homeostasis, impaired TGF-β signaling leads to hyperactive immune responses and contributes to various inflammatory diseases [144]. Thus, targeting the TGF-β pathway needs to be combined with anti-inflammatory therapies to reduce its side effects. Various strategies targeting TGF-β at the biosynthesis, activation, and signaling levels have been explored in fibrosis diseases, providing valuable insights for SS SG fibrosis treatment.

#### 4.2.1. Inhibiting TGF-β Activation

TGF-β bioactivity depends on ligand-receptor interaction, requiring activation to expose receptor-binding sites. Various factors can activate TGF-β, with integrin-dependent activation being the primary pathway. ROS, thrombospondin-1 (TSP-1), proteases, and other TGF-β activators mediate integrin-independent activation [144]. Targeting integrin-dependent pathways, studies have shown that blocking ανβ3 and ανβ5 integrin signaling can reduce TGF-β1 activation, inhibiting fibrosis in systemic sclerosis mouse models [145]. Additionally, the dual integrin αvβ1/αvβ6 inhibitor bexotegrast (PLN-78049) showed significant therapeutic effects in murine models of renal fibrosis, ameliorating organ damage and fibrosis [92]. A phase 2 clinical trial further confirmed that bexotegrast reduced lung fibrosis in idiopathic pulmonary fibrosis patients with good safety [93]. Although bexotegrast shows promise as a novel anti-fibrotic drug, its effects on other organs require further study.

TSP-1 is a key therapeutic target for integrin-independent activation. Studies have shown that the TSP-1 blocking peptide Leu-Ser-Lys-Leu (LSKL) inhibits TGF-β1 activation, preventing liver injury and fibrosis progression in a rat liver fibrosis model [94]. In a rat model of uremic dialysate injection, LSKL attenuated high glucose-mediated mesothelial-to-mesenchymal transition (MMT) and peritoneal fibrosis by blocking TSP-1 [95]. Although no studies have evaluated these drugs in SS, they provide valuable insights into the treatment of SS.

#### 4.2.2. Directly Targeting TGF-β

Small-molecule inhibitors or neutralizing antibodies directly targeting TGF-β have been used in fibrosis treatment. Fresolimumab, a high-affinity neutralizing antibody targeting all three TGF-β subtypes, showed significant therapeutic potential in clinical studies. In a phase 1 clinical trial on focal segmental glomerulosclerosis, fresolimumab demonstrated good safety and tolerability [96]. In an open-label clinical trial, the drug showed clear therapeutic effects in patients with systemic sclerosis, rapidly reversing skin fibrosis pathological markers and improving modified Rodnan skin scores (MRSS) [97]. However, the main adverse events of fresolimumab include skin disorders, bleeding, and anemia, with high-dose treatment in cancer patients leading to skin tumors, such as keratoacanthoma and basal cell carcinoma [98].

Pirfenidone, another important anti-fibrotic drug, has been shown in multiple clinical studies to inhibit TGF-β production and ECM deposition, effectively delaying fibrosis progression [99,100,101,102]. Due to its significant clinical efficacy, pirfenidone has been approved in multiple countries for the treatment of idiopathic pulmonary fibrosis and is a representative anti-fibrotic drug. However, its clinical application faces limitations; common adverse effects include gastrointestinal symptoms and rash [103], and some patients have discontinued treatment owing to intolerance, limiting its clinical use. Researchers have explored optimized dosing strategies, such as pirfenidone-loaded exosomes administered via tracheal instillation [104] and inhalable nanoemulsion [105,106], to reduce adverse effects and improve bioavailability and clinical efficacy. These strategies provide important insights for SS treatment, as local delivery can avoid interfering with TGF-β’s normal physiological functions and enhance therapeutic effects while reducing adverse effects. Therefore, exploring more precise local delivery methods in SS treatment has significant clinical implications.

#### 4.2.3. Targeting TGF-β Receptors

TGF-β signaling is mediated by TGF-β receptor I (TβRI) and TβRII, both enzyme-linked receptors with serine/threonine and tyrosine kinase activity, while TβRIII lacks direct TGF-β signaling motifs and primarily binds TGF-β2 with high affinity, acting as a co-receptor [144]. SB431542, a typical TβRI inhibitor, has been shown in multiple animal studies to inhibit ECM deposition, improve EMT, and suppress fibrosis progression in various diseases, such as retinal fibrosis, joint fibrosis, and liver fibrosis [107,108,109]. Another TβRI inhibitor, SM16, has been shown to prevent vascular fibrosis [110] and radiation-induced lung fibrosis [111], validating TβRI as a viable anti-fibrotic target. In addition to single-target TβRI inhibitors, GW788388, a compound inhibiting both TβRI and TβRII, has shown potential in the field of anti-fibrosis research. Studies have shown that GW788388 significantly inhibited kidney fibrosis in a diabetic nephropathy mouse model, indicating its broad application potential in the treatment of organ fibrosis [112]. Although TGF-β receptor inhibitors have shown significant efficacy in preclinical studies on various fibrotic diseases, further clinical trials are required to confirm their efficacy and safety. Therefore, future research should focus on conducting high-quality clinical trials to evaluate the practical applications of these inhibitors in SG fibrosis and other fibrotic diseases.

#### 4.2.4. Targeting SMAD Signaling

TGF-β signaling is activated through classical and non-classical pathways after binding to its receptors [144]. The classical TGF-β signaling pathway is primarily mediated by the SMAD transcription factor family, known as the SMAD signaling pathway. In this pathway, R-SMAD (including SMAD2 and SMAD3), are key mediators in tissue fibrosis and tumorigenesis, while I-SMAD (including SMAD6 and SMAD7) act as negative regulators, inhibiting excessive TGF-β signaling activation [67]. Based on this mechanism, inhibiting R-SMAD activation or enhancing I-SMAD function can precisely regulate TGF-β signaling.

Several small-molecule compounds have been developed to target the SMAD signaling pathway. For example, the alkaloid halofuginone (HT-100) effectively inhibits SMAD3 activation and reduces tissue fibrosis by decreasing the expression of MMP-2 and type I collagen. In multiple animal studies, halofuginone has been shown to improve fibrosis in various organs, including the skin, muscles, liver, and lungs [113,114,115]. However, Halofuginone induces significant toxicity to the liver and other human organs, thereby limiting its clinical application [116]. Additionally, the SMAD3-specific inhibitor SIS3 has shown good antifibrotic effects, alleviating renal and pulmonary fibrosis by inhibiting ECM remodeling [117,118]. In a mouse model of submandibular gland duct ligation, SIS3 blocked the TGF-β/SMAD3 signaling pathway and inflammatory factor expression, reducing acinar atrophy and ductal dilation, maintaining basement membrane morphology, and significantly decreasing interlobular and intralobular collagen deposition [119]. Notably, the G protein-coupled receptor kinase 2 (GRK2) inhibitor paroxetine has a unique anti-fibrotic potential. Studies have found that paroxetine inhibits Smad2/3 nuclear translocation, significantly improving glandular fibrosis, and delaying SS progression [120]. These studies suggest that targeting the SMAD signaling pathway may be a potential strategy for treating glandular fibrosis. Although existing studies have demonstrated its efficacy in animal models, additional preclinical and clinical trials are required to confirm its safety and efficacy.

### 4.3. Stem Cell Therapy

Mesenchymal stem cells (MSCs) exhibit both immunomodulatory and tissue repair capabilities and show positive therapeutic effects in SS. Studies have shown that MSCs can improve the salivary flow rates in NOD mice, reduce T- and B-lymphocyte infiltration in SGs, increase Treg cell numbers [121,123], decrease pro-inflammatory cytokine expression (IL-2, IL-4, IL-6, IL-17a, IFNγ, TNF-α), and increase anti-inflammatory cytokine IL-10 expression [124,125]. Additionally, MSCs reduce anti-SSA/Ro60 autoantibody production [126], further demonstrating their role in modulating autoimmune responses. In addition to immunomodulation, MSCs exhibit strong tissue repair capabilities. Through paracrine mechanisms, MSCs release various growth and trophic factors that promote SGEC proliferation and inhibit apoptosis in NOD mice [122,127]. After MSC treatment, SG acinar cell density increased, the acinar structure became more compact, and amylase levels significantly improved, indicating MSCs’ potential in repairing damaged SG tissue [128].

MSCs have shown good therapeutic efficacy in clinical studies. In a clinical trial involving 24 patients with SS, allogeneic MSC transplantation significantly reduced anti-SSA and anti-SSB antibody levels, decreased SS disease activity index (SSDAI) and visual analog scale (VAS) scores, and significantly improved clinical symptoms [129]. Additionally, MSCs play an important role in modulating the damaged ECM. MSCs protect the ECM from MMP-2 and MMP-9 degradation by secreting TIMP-1 and TIMP-2, thereby effectively improving SG fibrosis [130,131,132]. MSCs also alleviate inflammation through cell–cell contact, paracrine cytokines, and extracellular vesicles, contributing to SG ECM repair [133]. For instance, saliva-derived extracellular vesicles significantly attenuated TGF-β-induced fibrotic activity in human SG fibroblasts and SG organoids, inhibiting cell migration, collagen production, and myofibroblast activation. In in vivo experiments, local administration of these extracellular vesicles ameliorated SG fibrosis, protected acinar structure, and suppressed fibroblast STAT3 nuclear translocation [134].

Notably, MSCs and the ECM interact with each other. The direction of MSC differentiation is closely related to the local ECM. Studies have shown that MSCs can inhibit inflammation in early wound healing through TGF-β1 but may promote fibrosis by increasing myofibroblast differentiation in later remodeling stages [146]. Other studies have shown that MSCs cultured on pro-fibrotic ECM upregulate MMP expression and downregulate the key MMP inhibitors TIMP-1 and TIMP-2, indicating that MSCs in fibrotic ECM may exacerbate fibrosis by enhancing ECM degradation [147]. In a rat model, bone marrow MSCs transdifferentiated into SG epithelial cell lineages after induction by natural SG-specific ECM [148], confirming the efficacy of stem cell transplantation in SS treatment. This suggests that MSCs exposed to appropriate microenvironments during expansion can be induced toward specific phenotypes, minimizing phenotypic drift and maximizing their ability to restore damaged SG.

Not only does the ECM exert strong regulatory effects on MSCs, but MSCs also possess restorative capabilities toward the ECM. In an arthritis mouse model, extracellular vesicles (EVs) derived from human umbilical cord mesenchymal stem cells (hUC-MSCs) promote collagen type II production while suppressing the generation of MMP-1, MMP-3, and MMP-13 [149,150]. This results in enhanced synthesis and reduced degradation of the ECM. Furthermore, transplantation of MSCs into irradiated mouse SGs restored secretory function and attenuated glandular fibrosis [151].

In summary, MSCs exhibit anti-fibrotic properties by inhibiting inflammation and modulating ECM homeostasis; however, fibrotic ECM can also affect the MSC efficacy. Therefore, to maximize the therapeutic potential of MSCs, cell transplantation should occur in the relatively early stages of tissue damage when the local ECM is damaged but is still reversible, allowing MSCs to overcome the damaged SG ECM without adopting a pro-fibrotic phenotype. This strategy will enhance the efficacy of MSC treatment of SS and provide important theoretical insights for future clinical practice.

Although stem cell therapy holds promising prospects, it still faces limitations. These include safety risks (such as tumorigenicity and immune rejection), technical bottlenecks in precisely controlling cell differentiation and functional integration, challenges in large-scale production, and uncertainties in ethical regulations. Addressing these issues relies on advancements in technologies like gene editing, biomaterials, and tissue engineering, a deeper understanding of the disease microenvironment, and the establishment of more robust regulatory frameworks. Collaborative efforts from multiple stakeholders are required to overcome these hurdles.

### 4.4. Tissue Engineering Approaches

Tissue engineering and regenerative medicine show great potential for repairing and regenerating tissues and organs using engineered biomaterials and scaffolds. However, the complex biological properties and three-dimensional ultrastructure of natural ECMs make it challenging for traditional biomimetic scaffold systems to precisely replicate their specific structures and functional characteristics. Decellularized extracellular matrix (dECM) materials have emerged as promising solutions to this challenge. dECM is a biomaterial obtained by removing immunogenic cellular components from human or animal organs/tissues using decellularization techniques. This material is well known for its excellent biocompatibility, bioactivity, and mechanical properties. Importantly, dECM maximally retains the ultrastructure and composition of the natural ECM, providing an ideal environment and differentiation signals for implanted or recruited stem cells [135]. Stem cells actively participate in SG ECM repair by modulating inflammation, optimizing ECM components, and inhibiting fibrosis [130]. These two processes reinforce each other, forming a positive feedback loop.

In clinical practice, dECM are gradually being applied to support tissue repair and transplantation. Several dECM products are currently available on the market, such as Alloderm^®^, GraftJacket^®^, and Prima™ Plus, which are widely used for repairing skin, tendons, ligaments, and heart valves [152]. However, no commercial dECM products are available for the SGs. Studies have demonstrated its potential therapeutic value. Shin et al. successfully developed a dECM hydrogel derived from rat SG tissue (DSGM-hydrogel), and confirmed in vitro that this material provided a suitable microenvironment for stem cells and promoted their differentiation [136]. Additionally, Wang et al. prepared an injectable hydrogel derived from decellularized porcine submandibular glands (pDSG-gel) that significantly promoted SG MSC migration and recruitment by activating the PI3K/AKT signaling pathway. When pDSG gel was injected into the injury site of a rat SG defect model, researchers observed acinar and ductal regeneration, successfully inhibited fibrosis, and ultimately restored functional SG tissue [137]. These studies indicate that dECM materials have broad application prospects in SG repair. Although no commercial dECM products specifically for SGs are currently available, existing research lays a solid foundation for future clinical translation. With further research and technological advancements, dECM is expected to become an important tool for the treatment of SG injury and fibrosis.

Furthermore, in recent years, organoid technology as a crucial branch of regenerative medicine has achieved remarkable progress. Notably, organoids are not true human organs in the literal sense, but rather refer to three-dimensional (3D) structures derived from pluripotent stem cells or adult stem cells through in vitro cultivation. In the field of SGs, researchers have successfully induced branched tissue structures with glandular characteristics by cultivating embryonic SG tissues or pluripotent stem cells on compliant substrates containing specific ECM components. For example, Ogawa et al. integrated host salivary ducts with bioengineered SG ducts, implanting and developing bioengineered SG primordia in mouse models. This approach successfully formed correct tissue architectures and demonstrated potential for saliva secretion under neural stimulation [138]. In contrast, Sui et al. utilized FGF10 to induce human submandibular gland stem/progenitor cells to form induced organoid primordia. These were co-transplanted with mouse SG mesenchyme beneath the renal capsule of nude mice, where the cells responded to the mesenchymal niche to generate SG tissue exhibiting mature characteristics [139]. Notably, the study by Matsuno et al. successfully generated long-term expandable SG organoids using human induced pluripotent stem cell technology. After transplanting the resulting cell sheets into immunodeficient mouse models, these cells achieved partial integration with the host SG ducts and formed cross-species chimeric ducts. This result strongly demonstrates that organoid-based cell sheet transplantation represents a highly promising strategy for advancing functional SG regeneration [140].

However, one of the core challenges in current SG organoid research lies precisely in the difficulty of fully reconstructing the complex composition, 3D ultrastructure, and dynamic biological signaling networks of native SG ECM within in vitro culture systems. This deficiency in the ECM microenvironment leads to organoids’ inability to accurately replicate the highly sophisticated tissue architectures of in vivo glands—particularly regarding vascularization, neural networks, and complete ductal systems—while their secretory functions and responsiveness to physiological stimuli also significantly deviate from those of native glands. Consequently, exploring and integrating biomaterials that better mimic natural SG ECM properties (e.g., SG-derived dECM or its functionalized derivatives) as culture scaffolds or microenvironmental components represents a key strategy to overcome these limitations. Simultaneously, in vivo integration, functional restoration, and potential immunogenicity risks remain critical challenges [153,154]. These limitations not only compromise the structural integrity of SG organoids but also substantially restrict their post-transplantation survival and functional integration. Consequently, despite the successful in vitro generation of SG organoids, there remains a notable absence of successful cases demonstrating effective transplantation and therapeutic efficacy in disease treatment. Overcoming these limitations is a pivotal step in advancing SG organoids from valuable research models toward clinical applications.

### 4.5. Therapeutic Strategies

Current studies indicate that genetic, environmental, and immune factors collectively contribute to the pathogenesis and progression of SS, with immune dysregulation playing a central role. Breakdown of immune tolerance leads to systemic immune dysfunction, triggering aberrant activation and local infiltration of immune cells, which subsequently produce various autoantibodies that disrupt the structure and function of SGs. Under chronic exposure to pro-inflammatory mediators, SGECs undergo apoptosis or EMT, while the structure and composition of ECM become disordered, promoting continuous fibrotic progression. Therefore, anti-inflammatory and immunomodulatory therapies remain crucial strategies during the repair of SGECs and ECM. The combined application of anti-inflammatory treatment and ECM repair may enhance therapeutic efficacy through complementary mechanisms, achieving more comprehensive disease control. With the rapid advancement of tissue engineering therapies, combined treatment strategies are expected to bring new breakthroughs for SS management. For example, utilizing decellularized scaffolds to load or recruit stem cells can not only repair damaged cells but also improve the local microenvironment. Further combination with immunomodulatory drugs or other targeted agents may substantially enhance the treatment outcomes.

Beyond combination therapies, early intervention is equally crucial. During advanced fibrotic stages, severely disrupted tissue architecture often becomes irreversible and significantly compromises subsequent treatment efficacy. Early intervention not only effectively controls disease progression but also improves patients’ quality of life and reduces healthcare burdens. The key to translating this concept into clinical practice lies in identifying SS patients at high risk for fibrotic progression. Although no single gold-standard biomarker currently exists, integrating serological, imaging, and histopathological indicators can establish a multimodal risk assessment model. As previously mentioned, elevated MMP-3/TIMP-1 and MMP-9/TIMP-1 ratios in serum or saliva directly correlate with SG tissue damage and ECM remodeling; TGF-β1 serves as the core signaling molecule driving fibrosis; and high titers of anti-SSA/Ro52 antibodies may associate with more severe glandular damage and fibrotic predisposition [29,68,85]. These indicators collectively reflect the risk of fibrotic progression. Shear wave elastography, an ultrasound technique capable of quantitatively assessing SG stiffness, demonstrates increasing measurements that positively correlate with the degree of tissue fibrosis, providing a powerful tool for early fibrotic detection. Combining this technology with the OMERACT scoring system further enhances predictive accuracy [155,156]. Furthermore, labial gland biopsy remains the gold standard for SS diagnosis and can also serve as a risk assessment method. In addition to the conventional focus score, incorporating semi-quantitative fibrosis evaluation into pathological reports facilitates risk prediction [157].

Accordingly, we propose a preliminary clinical identification pathway. For patients diagnosed with SS, regular SG ultrasound examinations should be performed to monitor structural changes, alongside the detection of serum MMP-3/TIMP-1, MMP-9/TIMP-1 ratio, and TGF-β1 levels to assess molecular-level fibrotic driving signals. For patients showing structural abnormalities on ultrasound or significantly elevated serological markers, labial gland biopsy may be considered to determine the pathological fibrosis score. This “imaging + serology + pathology” multidimensional evaluation model can effectively identify patients at risk for advanced fibrosis, providing critical timing for implementing early targeted interventions.

## 5. Conclusions

In SS, the imbalance of the SG ECM—particularly the dysregulated expression of MMPs and the fibrotic process—has become a central link in understanding the disease’s pathological mechanism. This process interacts closely with chronic inflammation, cellular senescence, and the hyperosmolar microenvironment, collectively leading to the destruction of SG parenchymal structure and a progressive decline in secretory function. Based on these mechanisms, this article proposes several potential therapeutic strategies, including targeting MMPs and the TGF-β signaling pathway, as well as utilizing stem cells and tissue engineering technologies to promote repair.

Although ECM-targeted therapeutic strategies show promise, they still face numerous challenges. For instance, targeting MMPs and the TGF-β1 pathway requires enhanced specificity to avoid interfering with their physiological functions and reduce systemic side effects. Stem cell therapy possesses significant immunomodulatory and tissue repair capabilities, but its efficacy is considerably influenced by the timing of transplantation and the status of the local microenvironment. Furthermore, while tissue engineering approaches offer new possibilities for SG functional regeneration, technical bottlenecks remain in replicating the complex structure and function of the native gland.

Future effective ECM-targeted therapies will rely on the integration of early intervention and combined treatment strategies. Combining anti-inflammatory therapy with precise ECM modulation or integrating biomaterials with stem cell technology are promising directions worthy of exploration. These strategies are significant not only for SG repair but also provide valuable insights for repairing other tissue structures (such as skin and joints) and treating other autoimmune diseases.

## Figures and Tables

**Figure 1 biomolecules-16-00072-f001:**
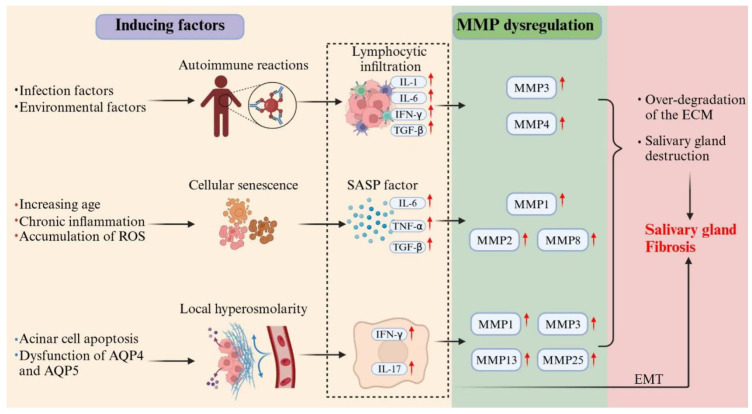
Multi-driver crosstalk in Sjögren’s syndrome SG fibrogenesis. This schematic illustrates the synergistic mechanisms driving SG fibrosis in SS. Autoimmune activation, cellular senescence, and hyperosmolarity collectively stimulate inflammatory cytokine production. These cytokines induce MMP dysregulation, leading to excessive ECM degradation and glandular architecture disruption. Concurrently, chronic inflammation promotes EMT and initiates aberrant scar-repair processes. The cycle culminates in progressive fibrotic tissue deposition, ultimately resulting in irreversible gland dysfunction.

**Figure 2 biomolecules-16-00072-f002:**
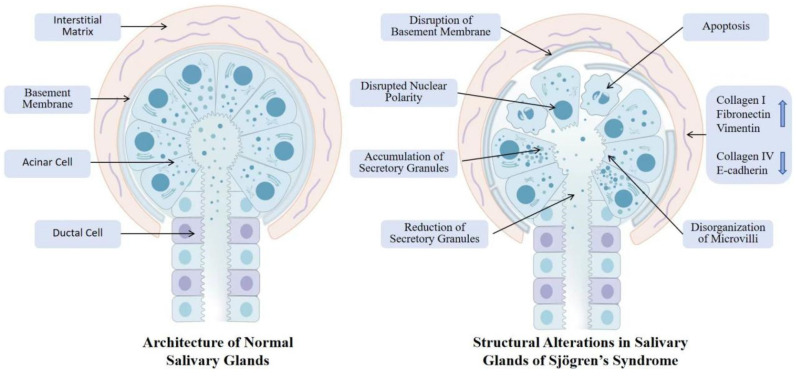
SG structural disruption in SS: normal vs. pathological features. This figure contrasts normal SGs architecture with pathological alterations in Sjögren’s syndrome. Normal SGs exhibit intact nuclear polarity, organized microvilli, continuous basement membrane, and balanced secretory granule dynamics within a structured ECM. In contrast, diseased glands display disrupted nuclear polarity, disorganized microvilli, increased apoptosis, aberrant accumulation or depletion of secretory granules, fragmented basement membrane, and ECM remodeling characterized by collagen deposition and laminin degradation. These structural defects impair secretory function, exacerbate inflammation–fibrosis crosstalk, and ultimately lead to irreversible gland dysfunction and xerostomia.

**Table 1 biomolecules-16-00072-t001:** Potential therapeutic approaches for repairing SG ECM.

Therapeutic Approach	Representative Drug/Material	Target	Research Stage	Efficacy and Challenges	References
Targeting MMPs	Harmine	MMP-3	In vitro experiments	Demonstrates antitumor and anti-inflammatory activities, but exhibits poor water solubility and dose-dependent toxicity	[86,87]
	Andecaliximab	MMP-9	Phase II/III clinical trial	Demonstrates favorable safety but lacks established therapeutic significance in Crohn’s disease and advanced gastric cancer	[88,89,90]
	JNJ0966	MMP-9	In vivo experiments	Exhibits high selectivity and ameliorates disease severity in experimental autoimmune encephalomyelitis mice	[78]
	(R)-ND-336	MMP-9	In vivo experiments	Exhibits high selectivity with anti-inflammatory and pro-angiogenic activities, accelerating wound healing in diabetic mice	[91]
Targeting TGF-β	Bexotegrast	αvβ1/αvβ6	Phase II clinical trial	Demonstrates a favorable safety profile, alleviates symptoms in idiopathic pulmonary fibrosis (IPF) patients, and inhibits fibrotic progression	[92,93]
	Leu-Ser-Lys-Leu (LSKL)	TSP-1	In vivo experiments	Inhibits hepatic and peritoneal fibrosis but lacks clinical evidence	[94,95]
	Fresolimumab	TGF-β	Phase I clinical trial	Ameliorates cutaneous fibrosis but induces dermatological disorders, hemorrhage, and anemia	[96,97,98]
	Pirfenidone	TGF-β	Marketing approval	Demonstrates confirmed efficacy while eliciting transient gastrointestinal symptoms and cutaneous eruptions in subsets of patients	[99,100,101,102,103,104,105,106]
	SB431542	TβRI	In vivo experiments	Inhibits retinal, articular, and hepatic fibrosis, but lacks clinical evidence	[107,108,109]
	SM16	TβRI	In vivo experiments	Prevents vascular fibrosis and radiation-induced pulmonary fibrosis, but lacks clinical evidence	[110,111]
	GW788388	TβRI/ II	In vivo experiments	Inhibits renal fibrosis but lacks clinical evidence	[112]
	Halofuginone	SMAD3	In vivo experiments	Demonstrates efficacy in ameliorating multiorgan fibrotic pathologies, but exhibits marked toxicities that constrain clinical translation	[113,114,115,116]
	SIS3	SMAD3	In vivo experiments	Inhibits renal and pulmonary fibrosis, and suppresses structural disruption of SGs in a submandibular gland duct ligation mouse model	[117,118,119]
	Paroxetine	SMAD2/3	In vivo experiments	Ameliorates glandular fibrosis and attenuates disease progression in a SS mouse model	[120]
Stem Cell Therapy	Stem Cell		Phase I/II clinical trial	Possesses dual immunomodulatory and tissue-reparative capacities that restore damaged SG epithelial cells and extracellular matrix, but incurs high production costs with inherent risks of immune rejection	[121,122,123,124,125,126,127,128,129,130,131,132,133,134]
Tissue Engineering Approaches	Decellularized extracellular matrix hydrogel		In vitro and in vivo experiments	Provides a native ECM microenvironment that enhances stem cell differentiation and tissue regeneration, suppresses fibrosis with functional restoration of SGs, though lacking commercially viable products targeting this specific organ	[135,136,137]
	Organoid		In vitro and in vivo experiments	Demonstrates great potential in modeling glandular functions and regenerative medicine, but still faces major challenges including immature structure/function and difficulties in integration with the body’s systems.	[138,139,140]

## Data Availability

Not applicable.

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
