# Peer review of "Biomolecules2026, 16(1), 72;https://doi.org/10.3390/biom16010072"

_biomolecules, 2026, doi:10.3390/biom16010072_

Round 1

Reviewer 1 Report

Comments and Suggestions for Authors

The article provides a comprehensive overview of the role of ECM and MMP dysregulation in Sjögren’s syndrome, highlighting the interplay between immune dysfunction, chronic inflammation, and fibrosis. The discussion of the challenges associated with broad-spectrum inhibition of MMPs and TGF-β is insightful and demonstrates a nuanced understanding of their physiological importance. The emphasis on early intervention and targeted therapies is particularly valuable, as it aligns with current trends in autoimmune disease management.

  1. Section 3 provides a detailed and well-structured discussion on the dysregulation of salivary gland (SG) matrix metalloproteinases (MMPs) while the coverage of SG fibrosis is relatively limited.
  2. The role of cytokines like IL-13, IL-17, and IL-22 in fibrosis is mentioned, but the mechanisms of IL-13 action are noted as unclear. Expanding on potential hypotheses or research directions for IL-13 would enhance the scientific rigor of the article.
  3. While the article mentions the lack of research focusing on the SG ECM, it could benefit from a deeper exploration of how ECM dysfunction specifically contributes to SS pathogenesis. For example, more detailed mechanisms or experimental findings related to ECM remodeling in SS would strengthen this section.
  4. The article highlights the importance of early intervention but does not elaborate on specific biomarkers or diagnostic criteria that could help identify patients at risk of late-stage fibrosis. Including such details would improve the practical applicability of the recommendations.
  5. The discussion on tissue engineering and organoid development is promising but remains somewhat general. Providing examples of specific studies or technologies that have shown success in repairing SGs or ECM in autoimmune diseases would add depth and credibility.

Author Response

We sincerely appreciate your valuable comments. In light of the extent of the revisions, we have provided detailed point-by-point responses in the attached file titled "Response to Reviewers' letter biomolecules-3974543.docx" for your convenience.

Reviewer 2 Report

Comments and Suggestions for Authors

In this review, Haodong Su and colleagues elucidate the characteristics of pathological and
physiological changes in the salivary glands (SGs), extracellular matrix (ECM) in Sjögren’s syndrome (SS), and thoroughly explored novel therapeutic strategies based on ECM regulation, and their clinical application prospects.

The manuscript is of clinical interest and addresses an important topic. However, to further improve the clinical/therapeutic impact, some issues should be addressed.

-Pathological Changes in SG ECM in Sjögren’s Syndrome: This is a very important topic discussing the fibrotic changes characterizing SS. However, the authors should recall important literature data demonstrating the similar histological features between SS and primary biliary cholangitis (PBC), a cholestatic chronic liver disease affecting the small intrahepatic biliary ducts. In particular, both PBC and Sjögren's syndrome are characterized by inflammation of target epithelial elements. Both diseases can be considered based on several other related clinical aspects, including proposed unique apoptotic features of the target tissue. PBC may be considered a Sjögren's syndrome of the liver, whereas SS can be equally discussed as PBC of the salivary glands, as previously suggested (doi: 10.1016/j.jaut.2011.11.005.). Importantly, from the serum autoantibody point of view, PBC and SS share a similar autoantibody profile characterized by the frequent detection of some antinuclear antibodies such as anti-SSA-Ro-R2kD in PBC patients, as previously demonstrated (DOI: 10.1111/j.1365-2036.2007.03433.x). This histological similarity might open novel anti-fibrotic treatments able to prevent disease progression, as anti-SSA-RO 52 kD antibodies were found associated with a more advanced histological stage of PBC (DOI: 10.1111/j.1365-2036.2007.03433.x).

Author Response

We extend our sincere gratitude to you for your valuable comments. For your convenience in reviewing, we have prepared a detailed response letter (Response to Reviewers' letter biomolecules-3974543.docx), and the specific responses can be found on pages 7-8 of this document.

Comment 1:

In this review, Haodong Su and colleagues elucidate the characteristics of pathological and physiological changes in the salivary glands (SGs), extracellular matrix (ECM) in Sjögren’s syndrome (SS), and thoroughly explored novel therapeutic strategies based on ECM regulation, and their clinical application prospects.The manuscript is of clinical interest and addresses an important topic. However, to further improve the clinical/therapeutic impact, some issues should be addressed.

Pathological Changes in SG ECM in Sjögren’s Syndrome: This is a very important topic discussing the fibrotic changes characterizing SS. However, the authors should recall important literature data demonstrating the similar histological features between SS and primary biliary cholangitis (PBC), a cholestatic chronic liver disease affecting the small intrahepatic biliary ducts. In particular, both PBC and Sjögren's syndrome are characterized by inflammation of target epithelial elements. Both diseases can be considered based on several other related clinical aspects, including proposed unique apoptotic features of the target tissue. PBC may be considered a Sjögren's syndrome of the liver, whereas SS can be equally discussed as PBC of the salivary glands, as previously suggested (doi: 10.1016/j.jaut.2011.11.005.). Importantly, from the serum autoantibody point of view, PBC and SS share a similar autoantibody profile characterized by the frequent detection of some antinuclear antibodies such as anti-SSA-Ro-R2kD in PBC patients, as previously demonstrated (DOI: 10.1111/j.1365-2036.2007.03433.x). This histological similarity might open novel anti-fibrotic treatments able to prevent disease progression, as anti-SSA-RO 52 kD antibodies were found associated with a more advanced histological stage of PBC (DOI: 10.1111/j.1365-2036.2007.03433.x).

Response 1:

We sincerely thank you for this highly insightful comment, which profoundly highlights the significant similarities between Sjögren's syndrome (SS) and Primary Biliary Cholangitis (PBC). We fully agree that this comparative perspective can substantially enhance the understanding of fibrotic mechanisms in SS and the exploration of therapeutic strategies.

In direct response to your suggestion, we have integrated this concept into our manuscript with a dedicated discussion in the revised Section 3.2 "Salivary Gland Fibrosis." The main additions are as follows:

  • Clarification of Histopathological Similarities: We have now explicitly described PBC as a disease that "mirrors" the pathological features of SS, characterized by the specific targeting of epithelial cells (bile duct epithelial cells vs. salivary gland duct/acinar cells), chronic inflammation, and progressive fibrosis. We have adopted the suggested analogy and cited the recommended literature, stating that "PBC may be considered 'Sjögren's syndrome of the liver,' while SS could be analogized as 'PBC of the salivary glands.'"
  • Integration of Shared Immunological Mechanisms: We have expanded the discussion to include the overlapping autoantibody profiles, particularly emphasizing the significance of anti-SSA/Ro52 antibodies. Crucially, we have cited the indicated study to highlight that the presence of these antibodies is associated with more advanced histological stages and greater fibrosis severity in PBC, thereby strengthening the argument for their potential role as effector molecules driving fibrotic progression in both diseases.
  • Proposal of Translational Research Directions: The added content concludes by emphasizing that in-depth investigation into the shared fibrotic pathways between SS and PBC may provide new perspectives and actionable targets for anti-fibrotic therapy in SS.

We are deeply grateful for this suggestion. By incorporating this comparative analysis, we believe our manuscript now offers a broader, more interconnected perspective on autoimmunity and fibrosis, significantly enhancing its conceptual framework and clinical relevance.

It should be noted that although SS and PBC exhibit significant similarities in their pathogenesis, there are important differences in their treatment strategies. Current anti-fibrotic approaches for PBC primarily focus on indirectly mitigating fibrosis by controlling disease activity (e.g., through choleretic and lipid-lowering agents), and its targeted therapies are mainly directed at liver-specific targets. Given this tissue specificity and the divergence in treatment strategies, we have not directly borrowed specific therapeutic regimens from PBC for SS in this revision. Instead, we have focused on exploring the common characteristics in the fibrotic pathways of both diseases from a mechanistic standpoint, thereby providing new directions for future basic research and target discovery.The specific changes are as follows:

It is noteworthy that the pathological process of SG fibrosis in SS exhibits similar-ities with another organ-specific autoimmune disease—Primary biliary cholangitis (PBC). PBC specifically targets the intrahepatic small bile duct epithelial cells, and its pathological features—chronic inflammation, bile duct destruction, and progressive liver fibrosis—closely mirror the damage to SG ducts and acini and the fibrotic patterns observed in SS. Consequently, some scholars suggest that PBC may be considered "Sjögren's syndrome of the liver", while SS could be analogized as "PBC of the salivary glands" [85]. This similarity extends beyond histology to immunological mechanisms. Both diseases share overlapping autoantibody profiles, with anti-SSA/Ro52 antibodies frequently detected in PBC patients. Notably, studies have shown that the presence of anti-Ro52 antibodies is associated with more advanced histological stages and greater fibrosis severity in PBC [86]. This suggests that in both SS and PBC, the immune re-sponse against common autoantigens like Ro52 may not only serve as a disease activity marker but could also be a key effector mechanism driving the progression from chronic tissue damage to fibrosis. Therefore, in-depth investigation into the shared fi-brotic pathways in SS and PBC may provide new perspectives for SS treatment.(It has been shown in the manuscript page8)

Reviewer 3 Report

Comments and Suggestions for Authors

Dear Authors, 

Please find attached comments. 

Author Response

We extend our sincere gratitude to you for your valuable comments. For your convenience in reviewing, we have prepared a detailed response letter (Response to Reviewers' letter biomolecules-3974543.docx), and the specific responses can be found on pages 9-10 of this document.

Comment 1: The manuscript should be clearly framed as a narrative review, and the title of the manuscriot should reflect the design.

Response 1:

Thank you for your valuable suggestion, and we also greatly appreciate your accurate grasp and summary of the core content of our article. We agree that clearly defining the type of manuscript will help readers better understand its positioning.

In accordance with your advice, we have explicitly defined the manuscript as a narrative review and accordingly updated the title. The new title is: "A Narrative Review of Salivary Gland Extracellular Matrix and Sjögren’s Syndrome: Research Status and Future Prospects"

We believe this revision accurately reflects the scope and nature of our article.

Comment 2: Conclusion could be rewriten to be shorter and more concise.

Response 2:

Thank you for your valuable suggestion, which made us realize that the original conclusion was not concise enough and lacked comprehensive summarization. We have thoroughly revised the conclusion to make it more concise and focused. The revised conclusion is now structured around three core aspects: systematically summarizing the key mechanisms of ECM dysregulation in SS and potential therapeutic strategies, objectively analyzing the current challenges, and clearly outlining future treatment prospects. This structural adjustment has enhanced the logical clarity and targeted expression of the conclusion. We sincerely appreciate your guidance.The specific changes are as follows:

In SS, the imbalance of the SG ECM—particularly the dysregulated expression of MMPs and the fibrotic process—has become a central link in understanding the dis-ease's pathological mechanism. This process interacts closely with chronic inflamma-tion, cellular senescence, and the hyperosmolar microenvironment, collectively leading to the destruction of SG parenchymal structure and a progressive decline in secretory function. Based on these mechanisms, this article prospects several potential therapeu-tic strategies, including targeting MMPs and the TGF-β signaling pathway, as well as utilizing stem cells and tissue engineering technologies to promote repair.

Although ECM-targeted therapeutic strategies show promise, they still face nu-merous challenges. For instance, targeting MMPs and the TGF-β1 pathway requires enhanced specificity to avoid interfering with their physiological functions and reduce systemic side effects. Stem cell therapy possesses significant immunomodulatory and tissue repair capabilities, but its efficacy is considerably influenced by the timing of transplantation and the status of the local microenvironment. Furthermore, while tis-sue engineering approaches offer new possibilities for SG functional regeneration, technical bottlenecks remain in replicating the complex structure and function of the native gland.

Future effective ECM-targeted therapies will rely on the integration of early in-tervention and combined treatment strategies. Combining anti-inflammatory therapy with precise ECM modulation, or integrating biomaterials with stem cell technology, are promising directions worthy of exploration. These strategies are significant not only for SG repair but also provide valuable insights for repairing other tissue structures (such as skin and joints) and treating other autoimmune diseases.(It has been shown in the manuscript page16)
